# Anonymous Asynchronous Ratchet Tree Protocol for Group Messaging [note 1]

**DOI:** 10.3390/s21041058

**Published:** 2021-02-04

**Authors:** Kaiming Chen, Jiageng Chen, Jixin Zhang

**Affiliations:** School of Computer, Central China Normal University, NO. 152 Luoyu Road, Wuhan 430079, China; cvk907@gmail.com (K.C.); zhangjixxx@foxmail.com (J.Z.)

**Keywords:** end-to-end encryption, forward secrecy, post-compromised security, anonymity, group messaging protocol

## Abstract

Signal is the first application that applies the double ratchet for its end-to-end encryption protocol. The core of the double ratchet protocol is then applied in WhatsApp, the most popular messaging application around the world. Asynchronous Ratchet Tree (ART) is extended from ratchet and Diffie-Hellman tree. It is the first group protocol that applies Forward Secrecy (FS) with Post-Compromised Security (PCS). However, it does not consider protecting the privacy of user identity. Therefore, it makes sense to provide anonymous features in the conditions of FS and PCS. In this paper, the concepts of Internal Group Anonymity (IGA) and External Group Anonymity (EGA) are formalized. On the basis of IGA and EGA, we develop the “Anonymous Asynchronous Ratchet Tree (AART)” to realize anonymity while preserving FS and PCS. Then, we prove that our AART meets the requirements of IGA and EGA as well as FS and PCS. Finally, the performance and related issues of AART are discussed.

## 1. Introduction

### 1.1. Background

With the help of Internet development, Instant Messaging (IM) applications are much important in people’s lives. According to statistics, WhatsApp is the most popular IM application around the world with more than 2 billion active users. Facebook Messenger has 1.3 billion users. The third is WeChat with about 1 billion. In 2018, people spent 27.6 h a week online, of which 15.6% was used for instant messaging. In addition, WeChat is the second IM application of China, and LINE is popular in East Asian countries. A large amount of data containing personal privacy information will be generated through these platforms.

End-to end encryption (E2EE) is used to protect user privacy such that the server or any attackers cannot read messages during the communication of IM. When the secret key is not compromised, Indistinguishability under Chosen Ciphertext Attack (IND-CCA) is considered as a standard to protect IM communication, in which case an attacker can request a prepared ciphertext [1]. However, when the secret key is compromised, there should be Forward Secrecy (FS) [2] and Post-Compromised Security (PCS) [3]. FS is to ensure that the adversary cannot obtain the key or plaintext information of the past secret messages. PCS is to guarantee that after multiple interactions, the compromised communication will be restored to a secure state again.

The group message protocol is extended from one-to-one IM with at least three users during the communication. The sender transmits a message, and the other group members will receive the corresponding one. Many protocols of IM applications directly send the message ciphertext, the encryption key, and the ciphertext of the key to each member with the one-to-one secure protocol. This strategy is called “sender keys”. Because the session key is determined by the sender, all members should keep the connection with others. This operation cannot meet the requirements of PCS because the receiver should obtain the identity of the sender to apply “sender keys”. To deal with this issue, ART protocol [4] is designed, which is based on the ideas of point-to-point [5] and stateful [3] protocols.

However, there are still issues. In 2019, WhatsApp was hacked through its phone call bug, which led to user information being leaked [6]. Thus, the user’s identity may be disclosed because of the engineering loopholes in the implementation of applications, and anonymous features are required. Current group message protocol cannot provide FS, PCS, as well as anonymity at the same time. Therefore, we aim to propose a protocol that can satisfy FS, PCS, and anonymity. According to the conference version [7], we re-formalize the two anonymous features, External Group Anonymity (EGA) and Internal Group Anonymity (IGA), as attack games to resist internal and external group attackers. In EGA, communications among different groups cannot be distinguished. Therefore, in EGA, attackers who are not members of a group cannot link users to the appropriate group. When the key is leaked, the external attacker can be regarded as a member of the group. EGA cannot resist such attackers. Therefore, IGA is required, in which other members cannot accurately locate the message sender except for the messages sent by themselves.

### 1.2. Contributions

In this paper, we develop the structure of ART to satisfy IGA security and apply the one-time address [8] to achieve the security of EGA. We formalize our construction with the algorithms Init to create the group channel, Enc to encrypt and send messages, and Dec to receive and decrypt messages. The sub-algorithm SKG is to derive the session key, and Update and UpdateGpk are to update group tree by sender and receiver, respectively. The tools are used to construct the following algorithms: a cipher $E=(E_{CPA},D_{CPA})$ which satisfies Indistinguishability under Chosen Plaintext Attack (IND-CPA), a MAC system I=(S,V) to protect the integrity of the message, and (Send,Get) to send and get messages from the server according to the one-time address. Then, we prove the security of AART that satisfies FS, PCS, and anonymity. Finally, we show that the performance of AART is better than the “sender keys” and pair-wise Signal group protocols and it is close to ART while providing anonymity features.

## 2. Related Works

In this section, we analyze the group protocols of IM applications and show that these protocols do not provide anonymity along with FS and PCS.

### 2.1. Group Protocols

#### 2.1.1. iMessage

Apple’s iMessage is the first popular E2EE application, but it turns out to be insecure under IND-CCA [9]. According to iMessage white paper [10], before sender A transmits a message to receiver B via iMessage, A should get the address of B from Apple’s server called APN because APN will store all users’ addresses. Furthermore, The group messaging protocol of iMessage is “sender keys”. Thus, anonymous features cannot be satisfied with iMessage.

#### 2.1.2. LINE

LINE [11] is an E2EE application that is popular in East Asia. According to the protocol of LINE called *Letter Sealing*, there are some issues such as impersonating attacks [12]. In group messaging, a group master key is calculated by the creator and sent to other members via “sender keys”. This master key will not be changed so that if it is compromised, the contents of communication will be revealed by the attacker. Thus, PCS is not satisfied in LINE.

#### 2.1.3. Signal

OTR [13] is the first application to provide ratchet. In ratchet protocol, users negotiate new Diffie-Hellman (DH) keys of each session, and the old session keys will be deleted and cannot be derived again. Signal’s protocol is called double ratchet. It proves that double ratchet can satisfy FS and PCS [5]. It can be observed from this protocol that long-term public keys are included in the associated data. So, the identities of users will be disclosed to the message server. Signal’s group messaging protocol is pair-wise, which requires that each member should maintain a one-to-one Signal protocol with other members rather than sending keys. Because of the pair-wise protocol, anonymity cannot be satisfied.

#### 2.1.4. ART

ART [4] is extended from ratchet and Diffie-Hellman tree, which first applies FS and PCS to group messaging protocol. The creator of a group generates DH key pairs for others. DH key pairs are set as the leaves of the DH tree, and the parents’ DH key pairs are generated from the ones of their children. The public DH tree is sent to other members. When sending messages, the sender needs to refresh his leaf DH key and the public DH keys from the corresponding leaf to the root of the group tree. The new public keys are sent to others to update their DH trees according to the location of the sender. Because the position of the sender is public and bound to the identity of the sender, ART cannot satisfy anonymity.

#### 2.1.5. WeChat and QQ

WeChat [14] and QQ [15] are the most popular IM applications in China. They apply Secure Sockets Layer (SSL) or Transport Layer Security (TLS) to protect the message security. A TLS connection will be set with the server to which the user is logged in. The group messages are transferred through this channel. TLS 1.3 proves to be FS. Though for PCS, because the later session key is derived from the former one, it cannot be satisfied in TLS, the same as WeChat and QQ. It is claimed that the identities of users can be protected. However, they do not offer technical details as well as the source code. Moreover, it is also not clear whether they are E2EE protocols or not.

### 2.2. Some Anonymous Approaches Applied in E2EE

Tor [16] is an anonymous network composed of many user volunteers. Tok [17] is the IM application based on Tor. When communicating, the sender randomly selects the same volunteer points, then derives long-time session keys of them. These keys are used to encrypt sending messages in sequence. According to the sequence, these messages are passed to the next point and decrypted by each point using the derived key until it is delivered to the receiver. Thus, the address of the sender is only known to the first point. This address of the receiver is only known to the last point. However, FS and PCS cannot be satisfied when the long-term keys are compromised.

Identity-based encryption (IBE) is used to validate and authenticate the anonymous public keys in E2EE [18]. Because of the low efficiency, KEM/DEM is applied to encrypt the secret key of the authenticator [19]. The encrypted secret key is sent to a proxy, then the proxy delays this message to the service provider for validation. As the proxy is trusted, the identity of the sender can be protected. Just like Tor, the secret key of the sender is long-term. So, it cannot provide FS and PCS.

## 3. Security Definitions

There are fundamental tools for the security definition. M is the message space. K is the key space. C is the cipher space. Σ is the MAC space. U is finite user identity set. E=(E,D) is the encryption scheme, E(k,m)=c:K×M→C is the encryption algorithm, and D(k,c)=m:K×C→M is the decryption algorithm. I=(S,V) is a MAC system where S(k,c)=σ:K×C→Σ and V(k,(c,σ))={0,1}:K×(C×Σ)→{0,1}. The output of *V* is 1 if a MAC pair is from *S*; if it is 0, *V* will reject this pair.

### 3.1. Algorithm Definition

The AART is the protocol with the following algorithms:$\left(gpk,gsk\right)\overset{\$}{\leftarrow }Init(\cdot$): it is the initialization algorithm to create group tree, generate the public group key gpk and public group key gsk.(C,σ)←Enc(gpk,gsk,m): it is the encryption algorithm to encrypt the message m with gpk and gsk. The outputs are a ciphertext *C* and a MAC σ.m∪⊥←Dec(gpk,gsk,C,σ): it is the decryption algorithm to check the σ and decrypt the ciphertext *C*. The output is the message *m* if it is decrypted correctly or ⊥ if it does not pass the validation of σ.

The sub-algorithms involved in the AART are defined as follows:$\left\{k_{1},...,k_{n}\right\}\leftarrow SKG\left(gpk,gsk\right)$: it is the session keys generation algorithm where $\left\{k_{1},...,k_{n}\right\}\in K^{n}$.(pos,path)←Update(gpk,gsk,pos): it is the update algorithm to refresh the leaf of the sender after he encrypts a message. pos is the position of the leaf to be updated, and path is the updated public key set in the group tree.gpk←UpdateGpk(gpk,pos,path): it is the update algorithm to replace part of the public keys of group tree according to path and pos after the receiver decrypts a message.

The encryption oracle Enc and decryption oracle Dec made up of these sub-algorithms and tools are illustrated in Figure 1.

### 3.2. Security Model

In the security models, messages queried by A are from M with the same length. In the challenge phase, the messages from A are different from queried messages. The adversaries mentioned in each definition are all probability polynomial time (PPT) attackers.

Unforgeability of MAC. The adversary on a MAC system attacks a chosen message and tries to forge a MAC pair that can pass the MAC system. The attacking game of unforgeability is shown in Figure 2. If $Adv_{UNF}=\left|Pr\left(V\left(k,m^{*},\sigma^{*}\right)=1\right)\right|$ is negligible, the MAC system can satisfy unforgeability.

Chosen Ciphertext Attack. The adversary of IND-CCA cannot only ask the plaintext encryption query but also has the ability to access decryption of the cipher. The attacking game of IND-CCA is shown in Figure 2. An encryption scheme S is IND-CCA if $Adv_{CCA}\left[A,S\right]=\left|\mathrm{Pr}\left(\hat{b}=b\right)-\frac{1}{2}\right|$ is negligible.

Forward Secrecy. The definition shows that the adversary cannot reveal the forward session keys when the keys are compromised. The attack game of FS is shown in Figure 2. Oracle O illustrates the forward encryption. After the challenge phase, the adversary can run decryption oracle Dec.

An encryption scheme S is FS if $Adv_{FS}\left[A,S\right]=\left|\mathrm{Pr}\left({\hat{b}}_{i}=b_{i}\right)-\frac{1}{2}\right|$ for any *i* is negligible.

Post-Compromised Secure. This definition shows that when the key is compromised after at most *Q* times queries, the channel will be refreshed and secure again. The attacking game of PCS is shown in Figure 2. The adversary can access the decryption oracle before and after the challenge phase. An encryption scheme S is PCS if $Adv_{PCS}\left[A,S\right]=\left|\mathrm{Pr}\left(\hat{b}=b\right)-\frac{1}{2}\right|$ is negligible.

Internal Group Anonymity. This definition shows that the adversary who knows the secret key cannot distinguish the identity of the target message sender. The attacking game of IGA is shown in Figure 3. An encryption scheme S is IGA secure if $Adv_{IGA}\left[A,S\right]=\left|\mathrm{Pr}\left(\hat{b}=b\right)-\frac{1}{2}\right|$ is negligible. After C receives the challenge, he should update the group tree according to the position *b*. If the Update algorithm of a protocol cannot cut off the relation between *b* and the updated position, the adversary will win the game. For the example of ART, because A knows the updated position of the sender, it means that in this definition, $c_{b,1}$ is related to *b* and can be accessed by A. So, in ART, $Adv_{IGA}\left[A,S\right]=1$.

External Group Anonymity. The security model of EGA is shown in Figure 3.

If $Adv_{EGA}\left[A,S\right]=\left|Pr\left(\hat{b}=b\right)-\frac{1}{2}\right|$ is negligible, an encryption scheme S is EGA. To make it indistinguishable, the only clue for the adversary is the output of Enc. It includes three parts: associated data pos and path, ciphertext *c*, and MAC σ. For ART and Signal, identity is an important associated data and easy to be distinguished. If an adversary cannot distinguish those associated data, it means that he cannot locate a user in an exact group.

## 4. Our Construction

### 4.1. Security Goals

Our construction aims to ensure security against the five kinds of adversaries in IND-CCA, FS, PCS, IGA, and EGA. All of the adversaries can deliver and modify the message, control the message server, and have the ability to access the decryption oracle. Except for IND-CCA, current random values including secret keys, session keys, and leaf keys can be compromised. To break the security features, the adversary can access the Key Derived Function (KDF) as a random oracle. Our construction does not consider the impersonating attack when the keys are compromised. Besides, the condition is not considered that the initial stage is compromised, and it assumes that the initial stage is based on a trusted third-party.

### 4.2. Security Assumption and Notation

In this subsection, the necessary assumptions and notations for AART are defined. $x\overset{\$}{\leftarrow }X$ means choosing a group element *x* from group *X* randomly. A secure pseudorandom generator (PRG) prg is to pick up the update position for group members. Sig is a secure signature, and I=(S,V) is a secure MAC system. $E=(E_{CPA},D_{CPA})$ is an IND-CPA encryption scheme, $Z_{q}$ is a finite field, *q* is a big prime number. The basic operation of AART is over point group P of Elliptic Curve (EC), where $P=\{\left(x,y\right)\in Z_{q}\times Z_{q}:\left(x,y\right)\in EC\}⋃\left\{\infty \right\}.$ The generator of P is *P*.

Decisional Diffie-Hellman Problem (DDHP). DDHP is to distinguish two tuples (a·P, b·P, ab·P) and (a·P, b·P, z·P), where $a,b\in Z_{q}$ and $z\overset{\$}{\leftarrow }Z_{q}$. The advantage for any PPT adversary to deal with DDHP is negligible.

Computational Diffie-Hellman Problem (CDHP). CDHP is to compute ab·P, given a tuple (a·P, b·P), where $a,b\in Z_{q}$. The advantage for any PPT adversary to deal with CDHP is negligible.

Pseudo-Random Function Oracle Diffie-Hellman (PRF-ODH) [20]. Assume a secure PRF t(·) is: $P\to Z_{q}$, which maps the group element of P to an element of $Z_{q}$. If DDHP is held in group P and *t* is a secure PRF over P, general PRF-ODH assumption is satisfied on P such that if $z\overset{\$}{\leftarrow }Z_{q}$, given (a·P,b·P,t(ab·P)),(a·P,b·P,t(z·P)), the probability adversary distinguishes t(ab·P), and t(z·P) is negligible. Because of PRF-ODH, CDHP is still satisfied over P and *t* if $z\overset{\$}{\leftarrow }Z_{q}$, given (a·P,b·P), the advantage that the adversary computes t(ab·P) is negligible.

Node.node is the basic unit of group tree. The construction of node is

node[i]: the *i*th leaf node of group tree;node[i].sk: the secret key of node[i];node[i].pk: the public key of node[i];node[i].sibling: the sibling of node[i];node[i].p: the parent of node[i].

Other operations are outlined: push is to push an element to the end of a list. pop is to get and remove the first element from a list. agt is the tree of public and private keys. size() is to get the number of group members or the number of a list. KeyExchange can be any authentication key exchange (AKE) function or protocol. In signal, KeyExchange is X3DH [5] protocol.
$$KeyExchange\left(ik_{R},IK_{I},suk_{R},EK_{I}\right)=KeyExchange\left(ik_{I},IK_{R},ek_{I},SUK_{R}\right)$$

This design involves several random values. The one-time secret key node[i].sk is owned to user *i*, node[i].pk is the corresponding public key. (ik,IK) is the identity key pair, (ek,EK) is the short-term key pair. ik and ek are kept by the user, and IK,EK are published. *j* denotes the sequence number of current stage. Session keys $mk_{j},r_{j},ck_{j}$ are derived from $KDF(ck_{j-1},tk_{j})$. $mk_{j}$ is used to encrypt message, $r_{j}$ is used to calculate one-time address, and $ck_{j}$ is used to generate MAC and session key pair for stage j+1.

### 4.3. Internal Group Anonymity

#### 4.3.1. Group Setup

Considering the three-member group, let A, B, and C be the group members. The initialization algorithm Init creates an anonymous group tree and sets up a communication channel. The leaves A, B, and C stand for each group member. This tree is created by the group initiator A. An overview of the group tree is shown in Figure 4.

The Init procedure is shown as follows:Ask for public key pairs ($IK_{i},EK_{i}$) of each group member through the third channel.Generate setup key $suk\overset{\$}{\leftarrow }Z_{q}^{*}$. Let SUK←suk·P. Generate A’s leaf key pair ($\theta_{0}^{A},\theta_{0}^{A}\cdot P$) such that $\theta_{0}^{A}\overset{\$}{\leftarrow }Z_{q}^{*}$. $\theta_{0}^{i}$ is the leaf secret key of user *i* and $\theta_{0}^{i}\cdot P$. Set initial chain key $ck_{0}\overset{\$}{\leftarrow }K$.Send $IK_{A},SUK,ck_{0}$ to other group members via a trusted third-party, which means that the adversary cannot access these messages and reveal the identity of other group members in the initial session.Generate leaf keys of other members: $\theta_{0}^{i}\leftarrow KeyExchange\left(ik_{A},IK_{i},suk,EK_{i}\right)$, generate random leaf key as $\theta_{0}^{i}\overset{\$}{\leftarrow }Z_{q}^{*}$.Set up group tree by agt←Create(). Let the root private key and public key be ($tk_{1},TK_{1}$). Set gpk as public group tree that deletes all secret keys from agt.Run $\sigma_{0}\leftarrow Sig\left(ik_{A},gpk_{1}\right)$ and broadcast $\left(gpk_{1},\sigma_{0}\right)$ to other group members.

Create and Init algorithms are illustrated in Algorithm 1.

When initiating anonymous group tree, the initiator has the full view of group tree, including the private leaf key of each node. After receiving this tree, other group members should check if $\left(IK_{A},gpk_{1},\sigma_{0}\right)$ is valid or not. If $\sigma_{0}$ is valid, each group member will accept this tuple. He will only obtain public part $gpk_{1}$ and his private leaf key. Leaf keys can be calculated by running
(1)$$\begin{array}{c} \theta_{0}^{i}\leftarrow KeyExchange\left(ik_{i},IK_{A},ek_{i},SUK\right) \end{array}$$

After getting $\theta_{0}^{i}$, group members should calculate their public leaf keys to ensure the position *i* of them. If the pk in $gpk_{1}$ of *k*th leaf is equal to $\theta_{0}^{i}\cdot P$, the position of this group member is i←k. Then, he generates the group shared key $tk_{1}$ according to procedure $KeyGen(i,node\left[i\right],gpk_{1}):$Parent node p←node[i].p,s←node[i]Find *s*’s sibling node s.siblingCalculate p.sk←t(s.sk·s.sibling.pk)set s←p,p←s.pIf *p* is null, tk←s.sk, else go to step 2

According to Equation (Equation 1), the group initiator knows the location of each member in $gpk_{1}$. However, each other member only knows his own location.
**Algorithm 1** Anonymous Tree Generation1:**function**Create (node,size)2:    **if**
size≠1
**then**3:        **if**
size is odd **then**4:           Let last node of newNode be node[size]5:        **end if**6:        **for**
i=1;i<size;i+=2
**do**7:           newNode[(i+1)/2].sk←t(node[i].sk·node[i+1].pk)8:           newNode[(i+1)/2].pk←newNode[(i+1)/2].sk·P9:           Let newNode[(i+1)/2] be the parent of node[i] and node[i+1]10:        **end for**11:        **return**
Create(newNode,size(newNode))12:    **else**13:        **return**
node14:    **end if**15:**end function**16:**procedure**Init($ik_{A},IK,EK$,size *n*)17:    $size\leftarrow 2n,suk\overset{\$}{\leftarrow }Z_{q}^{*},SUK\leftarrow suk\cdot P,ck_{0}\overset{\$}{\leftarrow }K$18:    Send $IK_{A},SUK,ck_{0}$ to other members through trust third-party19:    **for** each i∈[1,2n]
**do**20:        **if**
imod2=0 or i=A
**then**21:           $node\left[i\right].sk\overset{\$}{\leftarrow }Z_{q}^{*}$22:        **else**23:           $node\left[i\right].sk\leftarrow KeyExchange\left(ik_{A},IK_{i},suk,EK_{i}\right)$24:        **end if**25:    **end for**26:    agt←Create(node,size), gpk←agt, delete all sk from gpk27:    Run $\sigma_{0}\leftarrow Sig\left(ik_{A},gpk_{1}\right)$ and broadcast $\left(gpk_{1},\sigma_{0}\right)$ to other group members28:    **return**
gpk,agt,node29:**end procedure**

#### 4.3.2. Direct Updating

In order to satisfy FS and PCS, when one participant sends a message, the group tree should be updated. In stage *j*, the root key $tk_{j}$ should be generated from $gpk_{j}$ and the user’s leaf secret key. After sending or receiving a message, $gpk_{j}$ should be updated as $gpk_{j+1}$, which means that session key should be used only once. In the update phase, group members can decide to update the group tree anonymously or directly. The overview of directly updating is illustrated in Figure 5. Its procedure is described as follows (*B* stands for the position of the updated node):Set $node\left[B\right].sk\leftarrow \theta_{1}^{B}\overset{\$}{\leftarrow }Z_{q}^{*}$, node[B].pk←node[B].sk·PUpdate $sk_{2}\leftarrow t\left(\theta_{1}^{B}\theta_{1}^{y}\cdot P\right);pk_{2}\leftarrow sk_{2}\cdot P$Update $sk_{3}\leftarrow t\left(sk_{1}sk_{2}\cdot P\right);pk_{3}\leftarrow sk_{3}\cdot P$Update $tk\leftarrow t(sk_{3}sk_{4}\cdot P);TK\leftarrow tk\cdot P$Broadcast $B,node\left[B\right].pk,pk_{2},pk_{3}$ to all group members

After receiving the updated public keys, others update the public keys of *B* and its ancestor nodes, and $tk_{j+1}$ is derived according to KeyGen.

#### 4.3.3. Anonymous Updating

Because the group initiator knows the location of each member, he can see which one is to update group tree. So, the initiator knows who sent the target message. In order to limit the authority of the initiator, the relation between the updated location and identity should be separated. By using random node, this feature can be obtained according to Figure 6. The procedure is shown as follows (*b* stands for the updated node’s position):b←prg({2,4,6,...,2n})Set $node\left[b\right].sk\leftarrow \theta^{i}\overset{\$}{\leftarrow }Z_{q}^{*},node\left[B\right].pk\leftarrow node\left[B\right].sk\cdot P$Update $sk_{2}\leftarrow t\left(\theta_{1}^{B}\theta_{1}^{y}\cdot P\right);pk_{2}\leftarrow sk_{2}\cdot P$Update $sk_{3}\leftarrow t\left(sk_{1}sk_{2}\cdot P\right);pk_{3}\leftarrow sk_{3}\cdot P$Update $tk\leftarrow t(sk_{3}sk_{4}\cdot P);TK\leftarrow tk\cdot P$Broadcast $b,node\left[b\right].pk,pk_{2},pk_{3}$ to all group members

Because in group tree node[i],i∈{2,4,6,...,2n} are random nodes, this means that the leaf keys of these nodes are generated randomly, and thus no group member is located in these nodes. In this way, the initiator cannot bind the sender with a random node. Therefore, he cannot reveal the identity of the sender.

### 4.4. External Group Anonymous Encryption

#### 4.4.1. One-Time Address

Although ratchet tree can provide PCS and FS, it delivers messages through central servers. If those servers are controlled by the adversaries, they can know the relations of all users. With the help of the topological net, attackers can perform behavior analysis to infer the identities of the user.

One-time address applied in Monero [8] tries to hide the identity of receiver using Equation (Equation 2).
(2)$$\begin{array}{c} addr\leftarrow H\left(r\cdot PK_{B}^{s}\right)\cdot P+PK_{B}^{v} \end{array}$$

Here, $PK_{B}^{s}\leftarrow sk_{B}^{s}\cdot P$ and $PK_{B}^{v}\leftarrow sk_{B}^{s}\cdot P$ are the long-term public keys of user Bob. $H:P\leftarrow Z_{q}$ is a collision-resistant hash function. If user Alice wants to trade with Bob, she first generates $r\overset{\$}{\leftarrow }K$, calculates addr, and then puts r,addr and transactions onto the block chain. Bob should use *r* and his secret key pairs to validate the addr. Because addr is changed by *r* and *r* is randomly chosen, addr is changed in each transaction. Because DDHP is hard in PRF-ODH, the adversary cannot reveal the identity of Bob from addr. However, because Bob should check all addr, the valid operation will cost a lot of time. The idea from Monero’s one-time address is to hide the group public key, so that cloud servers cannot distinguish different messages from different groups according to one-time address. The SKG of our construction contains two parts: Equations (Equation 3) and (Equation 4).
(3)$$\begin{array}{c} mk_{j},r_{j},ck_{j}\leftarrow KDF\left(ck_{j-1},tk_{j}\right) \end{array}$$
(4)$$\begin{array}{c} addr_{j}\leftarrow H\left(t\left(r_{j}\cdot P\right)\right)\cdot P+tk_{j}\cdot P \end{array}$$

AART generates the pseudorandom value $mk_{j},r_{j},ck_{j}$ from $tk_{j}$ and $ck_{j-1}$ based on $KDF:K\times Z_{q}\to K^{3}$ modeled as random oracle, so that group members can pre-calculate the one-time address for each message.

#### 4.4.2. Encryption and Decryption

Here type∈{0,1} is the updated type: 0 is direct update, 1 is anonymous update.

$SKG(node{\left[i\right]}_{j},gpk_{j},ck_{j-1})$:–
$tk_{j}\leftarrow KeyGen\left(i,node{\left[i\right]}_{j},gpk_{j}\right)$
–
$mk_{j},r_{j},ck_{j}\leftarrow KDF\left(ck_{j-1},tk_{j}\right)$
–
$addr_{j}\leftarrow H\left(t\left(r_{j}\cdot P\right)\right)\cdot P+tk_{j}\cdot P$
$Enc(node{\left[i\right]}_{j},gpk_{j},type_{j},ck_{j-1})$:–
$\left(mk_{j},r_{j},addr_{j},ck_{j}\right)\leftarrow SKG\left(node{\left[i\right]}_{j},gpk_{j},ck_{j-1}\right)$

$\left(pos_{j},path_{j},gpk_{j+1}\right)\leftarrow Update\left(i,gpk_{j},type_{j},node{\left[i\right]}_{j}\right)$

$c_{j}\leftarrow E_{CPA}\left(mk_{j},m_{j}\right)$

$\sigma_{j}\leftarrow S\left(ck_{j},\left(c_{j},pos_{j},path_{j}\right)\right)$

$Send(\left(c_{j},pos_{j},path_{j},\sigma_{j}\right),addr_{j},server)$

$output:c_{j},\sigma_{j},addr_{j},gpk_{j+1}$

$Dec(gpk_{j},node\left[i\right],ck_{j-1})$
–
$\left(mk_{j},r_{j},addr_{j},ck_{j}\right)\leftarrow SKG\left(node{\left[i\right]}_{j},gpk_{j},ck_{j-1}\right)$
–
$cipher\leftarrow Get(addr_{j},server)$
–If cipher=⊥: output ⊥–
$c_{j},pos_{j},path_{j},\sigma_{j}\leftarrow cipher$
–If $V(ck_{j},\left(c_{j},pos_{j},path_{j}\right),\sigma_{j})\neq 1$: output ⊥–else: $\left(m_{j},pos_{j},path_{j}\right)=D_{CPA}\left(mk_{j},c_{j}\right)$–
$gpk_{j+1}\leftarrow UpdateGpk\left(pos_{j},path_{j},gpk_{j}\right)$
–
$output:m_{j},gpk_{j+1}$


Update is the algorithm to update the group tree during encryption, and UpdateGpk is to update the group tree after receiving updated path. The details of these two algorithms are illustrated in Algorithm 2. Send(msg,addr,server) means putting message msg on the server according to the position of addr. Get(addr,server) means getting the message from the position addr in the server. If sending is wrong or nothing is obtained, the response of the server is ⊥. These messages can be observed and accessed by the adversary.
**Algorithm 2** Update Group Tree1:**function**Update($i,gpk_{j},type_{j},node_{j}$)2:    if $type_{j}=0$, $pos_{j}=i$, otherwise $pos_{j}\leftarrow prg\left(\left\{2,4,6,...,2n\right\}\right)$3:    $node_{j+1}\leftarrow node_{j}$, $node{\left[pos_{j}\right]}_{j+1}.sk\overset{\$}{\leftarrow }Z_{q}^{*}$, $node{\left[pos_{j}\right]}_{j+1}.pk\leftarrow node{\left[pos_{j}\right]}_{j+1}.sk\cdot P$4:    **return**
pos,$UpdatePath(gpk_{j},node_{j+1},pos_{j})$5:**end function**6:**function**UpdatePath($gpk_{j},node_{j},pos_{j}$)7:    $cur\leftarrow node{\left[pos\right]}_{j+1},path_{j}\leftarrow \left[\;\;\right]$8:    **while** current node cur is not the root **do**9:        the sk of cur’s parent is t(cur.sk·cur.sibling.pk), the pk of cur’s parent is its sk·P10:        $path_{j}.push\left(cur.pk\right)$, let cur move to the parent of cur11:    **end while**12:    **return**
$path_{j},cur$13:**end function**14:**function**UpdateGpk($pos_{j},gpk_{j},path_{j},node_{j}$)15:    $tmp\leftarrow node[pos_{j}]$16:    **while**
$path_{j}\neq \left[\;\;\right]$
**do**17:        $tmp.pk\leftarrow path_{j}.pop\left(\right)$, tmp←tmp.p18:    **end while**19:    **return**
tmp20:**end function**

## 5. Security Analysis

In this section, it proves that AART satisfies the secure definitions of IND-CCA, FS, PCS, IGA, and EGA. The sequence of current stage is *j*.

### 5.1. IND-CCA Security

**Theorem** **1.**
*Let $E\leftarrow (E_{CPA},D_{CPA})$ be a cipher, and I←(S,V) is a MAC system. $KDF:K\times Z_{q}\to K^{3}$ is modeled as a random oracle. Assuming E is IND-CPA secure and I is a secure MAC system, if adversary A has the advantage to break IND-CCA of AART, with $Q_{d}$ times decryption queries and $Q_{H}$ time Random Oracle queries, then there exists an adversary $B_{UNF}$ against I, an adversary $B_{PRF-ODH}$ against CDHP in PRF-ODH, and an adversary $B_{CPA}$ against IND-CPA of E with the following bound:*
(5)$$\begin{array}{cc} Adv_{CCA}^{RO}\left[A,AART\right]\leq & Q_{H}\cdot Adv_{CDHP}\left[B_{PRF-ODH},P\right]+Adv_{CPA}\left[B_{CPA},E\right]\\ & +Q_{d}Adv_{UNF}\left[B_{UNF},I\right] \end{array}$$


**Proof.** In each Game${}_{j}$, *b* is randomly chosen by C, and $\hat{b}$ is the output of A. $W_{j}$ is the event that in Game${}_{j}$, $b=\hat{b}$. The decryption query is defined in Game${}_{0}$ as
When receiving $c_{j},\sigma_{j}$ from adversary, check if $V(k_{0},c_{j},\sigma_{j})=1$.If it is true, reply $D(k_{1},c_{j})$, else ⊥.It should prove that
(6)$$\begin{array}{c} Adv_{CCA}^{RO}\left[A,AART\right]=\left|\mathrm{Pr}\left(W_{0}\right)-\frac{1}{2}\right| \end{array}$$Then, Game${}_{0}$ is changed into Game${}_{1}$. Step 1 is deleted and step 2 is changed to send “reject” except when $j=\omega \in \{1,Q_{d}\}$. It can be seen that the difference between Game${}_{0}$ and Game${}_{1}$ is the event that $c_{\omega }$ is queried. According to the definition of Unforgeability, there is
(7)$$\begin{array}{c} Adv_{UNF}\left[B_{UNF},I\right]=\left|\mathrm{Pr}\left(W_{0}\right)-\mathrm{Pr}\left(W_{1}\right)\right|/Q_{d} \end{array}$$To simplify, we will remove the decryption query in accordance with Equation (Equation 7) from our proofs. Thus, Game${}_{1}$ is the IND-CPA game of AART and then is modified into Game${}_{2}$.The random oracle is recorded by MAP. Game${}_{2}$ is the same as Game${}_{1}$ except for deleting MAP operation of step 8 from Game${}_{1}$. Event Z is defined such that A queries $tk_{Q_{1}+1},ck_{Q_{1}+1}$ in domain(MAP). The difference between these two games is that event Z happens. So there is
(8)$$\begin{array}{c} |\mathrm{Pr}\left(W_{2}\right)-\mathrm{Pr}\left(W_{1}\right)|=\mathrm{Pr}\left(Z\right) \end{array}$$**Using**CDHP. If event Z happens, it means that A queries $tk_{Q_{1}+1},ck_{Q_{1}+1}\in$ domain (MAP), which can be used to break CDHP and to construct $B_{PRF-ODH}$. To break CDHP, one tk,ck pair should be picked out, but $B_{PRF-ODH}$ is not sure which one in domain(MAP) is the right answer. Assume there are at most $Q_{2}$ times random oracle queries; the probability to select right pair is at most $\frac{\mathrm{Pr}(Z)}{Q_{2}}$. We use Game${}_{2}$ to construct Game${}_{CDHP}$. Instead of running Init,KeyGen,Update, $B_{PRF-ODH}$ should query them from $C_{PRF-ODH}$. The gray parts with boxes of Game${}_{2}$ challenger are constructed as $C_{PRF-ODH}$. Thus, from A’s view, there is no difference between Game${}_{2}$ and Game${}_{CDHP}$. Event Z happens ⇔$tk_{Q_{1}},ck_{Q_{1}}\in$ domain(MAP) when $B_{PRF-ODH}$ finishes the game. Let $Q\leftarrow Q_{2}$, because the pairs may be queried more than once, the size of domain(MAP) is no greater than *Q*. So, there is
(9)$$\begin{array}{c} Adv_{CDHP}\left[B_{PRF-ODH},P\right]\geq \frac{\mathrm{Pr}(Z)}{Q} \end{array}$$According to Game${}_{2}$, to deal with $\mathrm{Pr}(W_{2})$ means to deal with IND-CPA. So
(10)$$\begin{array}{c} |\mathrm{Pr}\left(W_{2}\right)-\frac{1}{2}|=Adv_{CPA}\left[A,AART\right] \end{array}$$**Using CPA**. Game${}_{CPA}$ can be constructed from Game${}_{2}$. Let Game${}_{2}$ challenger be $B_{CPA}$ except that after receiving message from A, $B_{CPA}$ should run encryption query to $C_{CPA}$ such like the gray parts with no boxes in Figure 7. So there is
(11)$$\begin{array}{c} Adv_{CPA}\left[A_{CPA},AART\right]=Adv_{CPA}\left[B_{CPA},E\right] \end{array}$$Combining Equations (Equation 6)–(Equation 11), Theorem 1 can be derived. Because CDHP in PRF-ODH is hard and *E* is IND-CPA cipher, *I* is secure MAC system, A cannot win Game${}_{0}$. So $Adv_{CCA}\left[A,AART\right]$ is negligible. IND-CCA of AART is satisfied. □

### 5.2. Forward Secrecy

**Theorem** **2.**
*Let $KDF:K\times Z_{q}\to K^{3}$ be modeled as a random oracle. When the keys of stage j+1 are leaked, if adversary A can break FS of AART, there exists adversary $B_{CCA}$ that can break the IND-CCA of stage j with the advantage:*
(12)$$\begin{array}{c} Adv_{FS}^{RO}\left[A,AART\right]\leq Q\cdot Adv_{CCA}^{RO}\left[A,AART\right] \end{array}$$


**Proof.** Assume there are *Q* stages. According to SKG and Update, $tk_{j}$ is derived from $gpk_{j}$, and session keys of stage *j* are generated by $tk_{j},ck_{j-1}$. So if all random values including sk of each user, $tk_{j}$, session keys $mk_{j},r_{j},ck_{j}$ are compromised, and adversary A wants to get session keys of stage j−1, he needs to know $tk_{j-1}$. If the current leaf key of each user is not compromised, each stage can be reduced to an IND-CCA game in Theorem 1. If the current leaf key is compromised, he can get $tk_{j-1}$ when the leaf key is not updated. So he can try to get $ck_{j-2}$ to break FS. In order to get $ck_{j-2}$, he should get $ck_{j-3}$ recursively until the initial stage. However, the initial stage is run through secure AKE and a trusted third-party, and the adversary cannot break FS through this way. Assume challenger C is the group creator. Game${}_{0}$ is illustrated in Figure 8.For the *i*th message query, if ${\hat{b}}_{i}=b_{i}$, A wins Game${}_{0}$. By querying each session key, root key, and plaintext encryption from the IND-CCA challenger of Game${}_{0}$ in Figure 7, Game${}_{0}$ can be changed into Game${}_{CCA,i}$ for each stage *i*. According to Theorem 1:
(13)$$\begin{array}{c} Adv_{FS}^{RO}\left[B_{FS,i},s_{i}\right]\leq Adv_{CCA}^{RO}\left[A,AART\right] \end{array}$$There are *Q* times of Game${}_{i}$, so Theorem 2 proves to be true. Because $Adv_{CCA}^{RO}\left[A,AART\right]$ is negligible, $Adv_{FS}^{RO}\left[A,AART\right]$ is negligible too. Forward Secrecy of AART is satisfied. □

### 5.3. Post-Compromised Security

PCS is proved with Theorem 3.

**Theorem** **3.**
*Let $KDF:K\times Z_{q}\to K^{3}$ be modeled as a random oracle. When the keys of stage j are compromised, if in the challenge stage all leaf keys are updated, the advantage of adversary A to break PCS of AART is equal to the advantage of A to break IND-CCA of stage j+1, such that*
(14)$$\begin{array}{c} Adv_{PCS}^{RO}\left[A,AART\right]=Adv_{CCA,j+1}^{RO}\left[A,AART\right] \end{array}$$


**Proof.** When other keys except for $ck_{j}$ of *j*th session are compromised, because the keys of the next session j+1 are based on $ck_{j}$, the adversary cannot derive them. So, the only way for the adversary is to break the IND-CCA of j+1 session. Thus, Theorem 3 can be reduced. When all keys are compromised, if the leaf keys adversary holds are not updated until the *Q* session finished, the advantage for the adversary is 1. However, when each leaf key of the group tree is updated, the advantage of A is reduced to the IND-CCA of *Q*th session and becomes negligible. □

### 5.4. Internal Group Anonymity

IGA of AART is proven with Theorem 4.

**Theorem** **4.**
*Let KDF be modeled as random oracle, $E_{CPA}$ be IND-CPA cipher, and prg be secure PRG; if there exists adversary A to break IGA, then there exists adversary B that breaks PRG:*
(15)$$\begin{array}{c} Adv_{IGA}\left[A,AART\right]=\left|\mathrm{Pr}\left(b=0\right)-\mathrm{Pr}\left(\hat{b}=0\right)\right|=Adv_{PRG}\left[B,prg\right] \end{array}$$


**Proof.** Because the random leaf to be used in the anonymous update is chosen randomly by secure PRG, if the adversary can distinguish between two anonymous users from each other depending on their updated messages, he can break the security of PRG. □

### 5.5. External Group Anonymity

**Theorem** **5.**
*Let H be a collision-resistant hash function and KDF be modeled as random oracle; if adversary A can break EGA of AART, there exists adversary $B_{PRF-ODH}$ against DDHP in PRF-ODH with the advantage:*
(16)$$\begin{array}{c} Adv_{EGA}\left[A,AART\right]\leq 2\cdot Adv_{DDHP}\left[B_{PRF-ODH},P\right] \end{array}$$


**Proof.** Illustrated as Figure 9, Game${}_{EGA}$ includes two parts Game${}_{0}\left(0\right)$ and Game${}_{0}\left(1\right)$ simulating two groups. Challenger C plays Game${}_{0}\left(b\right)$ with adversary A where $b\overset{\$}{\leftarrow }\left\{0,1\right\}$. A should distinguish which game is played. If the output of A is $\hat{b}$ and $\hat{b}=b$, A win Game${}_{EGA}$. For each Game${}_{0}\left(b\right)$, a DDHP game can be constructed such that $tk_{b}$ is generated from random as Game${}_{1}\left(b\right)$. $W_{0}^{b}$ denotes that Game${}_{0}\left(b\right)$ is played and $W_{1}^{b}$ denotes that Game${}_{1}\left(b\right)$ is played. According to the definition of EGA, there is
(17)$$\begin{array}{c} Adv_{EGA}\left[A,AART\right]=\left|\mathrm{Pr}\left(W_{0}^{0}\right)-\mathrm{Pr}\left(W_{0}^{1}\right)\right| \end{array}$$According to the definition of DDHP in PRF-ODH, there is
(18)$$\begin{array}{cc} Adv_{DDHP}\left[B_{PRF-ODH},P\right] & =|\mathrm{Pr}\left(W_{0}^{b}\right)-\mathrm{Pr}\left(W_{1}^{b}\right)|=|\mathrm{Pr}\left(W_{0}^{b}\right)-\frac{1}{2}|\\ 2Adv_{DDHP}\left[B_{PRF-ODH},P\right]\; & \geq |\mathrm{Pr}\left(W_{0}^{0}\right)-\mathrm{Pr}\left(W_{0}^{1}\right)|=Adv_{EGA}\left[A,AART\right] \end{array}$$Then, Theorem 5 proves to be true. Because DDHP is hard in PRF-ODH, $Adv_{EGA}$ is negligible. So EGA of AART is satisfied. □

## 6. Discussion

We further discuss the performance and some issues when running AART.

Performance. The performance comparison can be seen from Table 1. For *n* group members, the number of nodes of ART is 2n. The amount of nodes in AART is 4n because of the additional random nodes. Thus, the exponentiation times and storage cost to generate the public tree of AART are two times as ART. Also, the height of the group tree will be log(2n)+1 in AART, which is increased by one compared with log(n)+1 in ART. The complexity and storage in update phase will retain the same relationship of the heights. Moreover, there is an additional addr in AART. Above all, the complexity and storage of AART are close to ART.

For the exponentiation times, it will be 4n for the sender in AART because of the tree structure. Because of the Update algorithm, the time cost in the following stage will be log(2n). The sender of the pair-wise Signal should update all of the channels with others. Thus, it will cost *n*, worse than AART. “Sender keys” will not refresh their channels, it will be 0.

For encryption times, only “sender keys” will encrypt the message keys for others. For all of these protocols, there will be only one encryption operation in each stage.

For communication storage, the sender of AART should store the n−1 long-term public keys of others and broadcast the 4n public key pairs to others; it will be 5n−1. Each group member should not know the long-term public keys of other group members except for the creator, the cost will be 4n+1. In ART, each member should get the identity keys of others. The ongoing cost will be log(2n) because of the outputs of the Update operation. “Sender keys” will cost *n* for sending keys at the beginning, but it is only 1 ongoing since the ciphertext for each member is the same. According to one-to-all channels, it will take up *n* for both sender and others through pair-wise Signal. In the following sessions, it will cost *n* to refresh all channels between the sender and receivers. The computation storage is the addition of storage spent on exponentiation and encryption. It can be seen that the cost of AART at the setup stage is the largest. However, because of the tree structure, AART is more efficient in the ongoing stages compared with pair-wise Signal.

Although iMessage provides E2EE features, it cannot resist against the CCA [9] level attacker. LINE applies E2EE, but it cannot achieve FS and PCS. Tor is not an E2EE protocol because the last node of Tor knows the plaintext of the sender. ART is the first group protocol applying PCS, but it cannot cope with identity protection. With the help of the additional cost, AART can achieve FS, PCS, and anonymity at the same time compared with other protocols. The security comparison can be seen in Table 2.

About trusted third-parties. In ART, there is no efficient way to protect the initial stage from being attack. If the first is compromised, it means that all of the users’ long-term secret keys can be access, and the identities of group members will be obtained at the beginning. We follow this setting, and we initialize the first stage session key by $tk_{1}$ and $ck_{0}$. The later $ck_{j}$ is generated by former root key $tk_{j}$ and $ck_{j-1}$. Thus, the $ck_{0}$ should be either empty or decided by the group creator. If $ck_{0}$ is empty, the FS cannot be satisfied when $tk_{j}$ is compromised. The details can be found in the proof of Theorem 2.

Anonymity when the key is compromised. From Theorems 2 and 3, AART can provide FS and PCS. However, it should be considered whether AART will still satisfy IGA and EGA when the key is compromised. In IGA, the adversary can be seen as the group creator, according to Theorem 4, the adversary cannot distinguish the identities of the senders even when he knows all of their secret keys. For the EGA adversary, if the key is compromised, he may know the identities when the identity keys are leaked. According to Theorem 5, he at least cannot reveal who sends the target message.

IP address. Message server may bind the IP addresses with users who access the same addr in the server. To avoid this situation, users can visit the server through a proxy. According to Tok, the out point of Tok should know the IP address of users. This situation cannot be avoided. However, AART just concerns the addr in the server. If the proxy is not controlled by the adversary and message server, or the proxy IP address is changed all the time, the adversary cannot bind the IP of users with the same group. Thus, the adversary cannot reveal the real relation of the group members in the real world.

Message conflict. In the real network environment, group members may send messages at the same time but generate different gpk of next stage, which will cause conflict and break the protocol. In AART however, all users of the same group will generate the same addr. If addr exists, it means that the updating operation is out of date and the message should be re-encrypted again. To avoid the adversary or server taking up the addr of the current stage, the sender can check the MAC of addr. If it is wrong, this addr is still available for the group. When the key is compromised, AART cannot avoid the situation that the adversary generates the same addr and legal MAC value. However, this ability belongs to the active attacker, and we aim to prevent the adversary to become an active attacker.

Message recovery and chosen ciphertext attack. To recover messages of a group, group members should keep their sk of all stages along with the initial $gpk_{1}$. According to $gpk_{1}$ and $sk_{1}$, users can generate $addr_{1}$ and get the correct message matched by $addr_{1}$. Thus, users can update the correct $gpk_{2}$, while they also hold $sk_{2}$. That means all messages can be put in the server and can be recovered correctly.

Keeping all sk will weaken the security of AART. For IND-CCA, the challenge can reject the decryption query because the structure of AART is also a ratchet and can only be pushed forward but not in the backward direction. So, users only own the secret key of the current stage, ideally, and the former stage for the consideration of message conflict. Besides the definitions that the adversary cannot inquire about the messages in plaintext query, if the adversary can access old information and ask for decryption, the challenge will reject this request because the MAC key of old information has been deleted, and the probability that these two keys are the same is negligible. However, if users store the past secret keys, the challenge should set up a table to combine the secret keys with old messages. When queried by the IND-CCA adversary in this situation, the challenge should look for the table and decrypt the message if the requested message is matched. Therefore, in a message recovery situation, AART cannot resist the IND-CCA adversary. If IND-CCA is required, the message recovery should be given up.

Malicious group member. Malicious users who want to compromise keys or combine two group trees are included in without the help of the leaked keys. For the former situation, because of FS, PCS, IGA, and EGA, messages, as well as identities, can be protected. For the latter, although a malicious user can replace his leaf key in group A with the root of another group B, since the chain keys are different in two groups, members of group A cannot get the addr of B. Therefore, the two groups cannot be combined.

About collusion attacks, in a group of *n* members, if there are n−1 members in collusion, including the creator and the rest sending a message, they can reveal the identity of him. However, if the creator is trustworthy, collusion attackers can only know that one member sent a message, but they cannot reveal the identity of him because they do not know the long-term public key of the sender.

Dynamic group member and device. It is easy to add a new group member through KeyExchange. The initial leaf key can be obtained by the creator, and then the creator creates a three-node agt with one root, a new member leaf, and a new random leaf. Then the creator inserts the three-node agt to the current agt to be a complete binary tree (two leaves and their parent are thought to be one unit). The creator uses tk and three nodes agt’s root public key to generate new agt’s root tk and public TK. Finally, he publishes the new gpk of agt. Deleting a member is easy as well. Consider one unit as a three-node agt including a user leaf, its sibling random leaf, and their parent node, the sibling of one unit has the same parent node with this unit. To remove one user, the creator should replace the parent of the unit where the target user is located with the sibling unit, use the random leaf in the sibling unit to update the agt, and publish the new gpk to all group members.

Regarding the dynamic device, the user can share tk and ck with multiple devices, create a subtree, and let the root of the subtree replace the user leaf. When updating agt, the user should update this subtree and output the path except for the path in this subtree to group members. Then, other group members will believe that they are chatting with a multi-device user.

## 7. Conclusions

In this paper, we propose a multi-stage anonymous group messaging protocol called AART, which is based on the design of ART. It can provide anonymity features including IGA and EGA, while it retains the previous features such as FS and PCS of ART. The security of AART is analyzed formally. Finally, we discuss the performance of AART compared with ART, pair-wise Signal, and “sender keys” protocols as well as other problems that may exist in AART and the related solutions to them. In our future work, the effort will be focused on how to limit anonymity by tracing the secret keys and revealing the identity of malicious users.

## Figures and Tables

**Figure 1 sensors-21-01058-f001:**
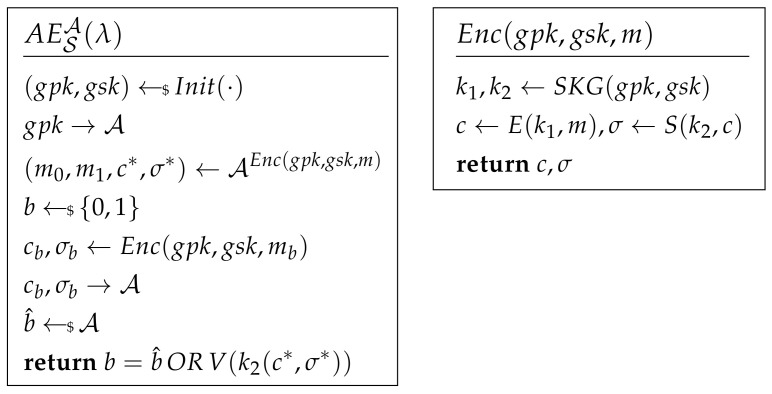
Authenticated encryption.

**Figure 2 sensors-21-01058-f002:**
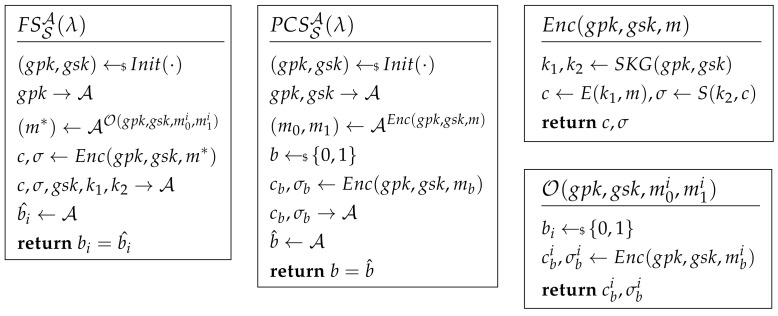
Forward Secrecy and Post-Compromised Security.

**Figure 3 sensors-21-01058-f003:**
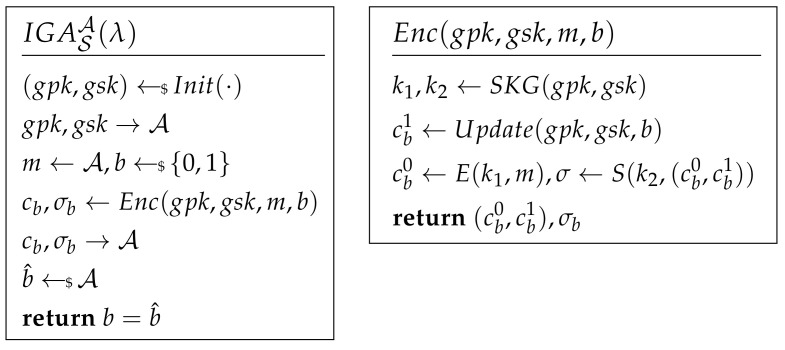
External Group Anonymity.

**Figure 4 sensors-21-01058-f004:**
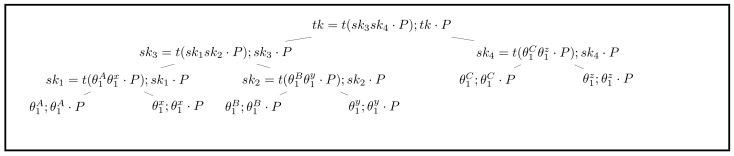
Anonymous group tree overview.

**Figure 5 sensors-21-01058-f005:**
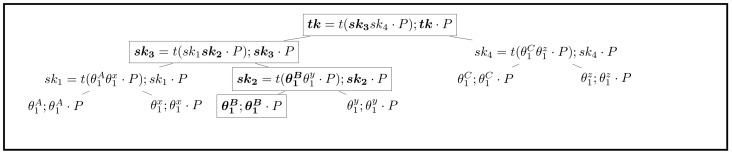
Non-anonymous updating group tree (updated nodes and values are marked in bold).

**Figure 6 sensors-21-01058-f006:**
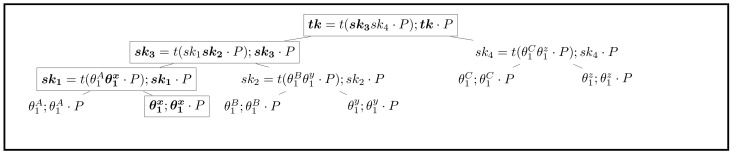
Anonymous updating group tree (updated values are marked in bold).

**Figure 7 sensors-21-01058-f007:**
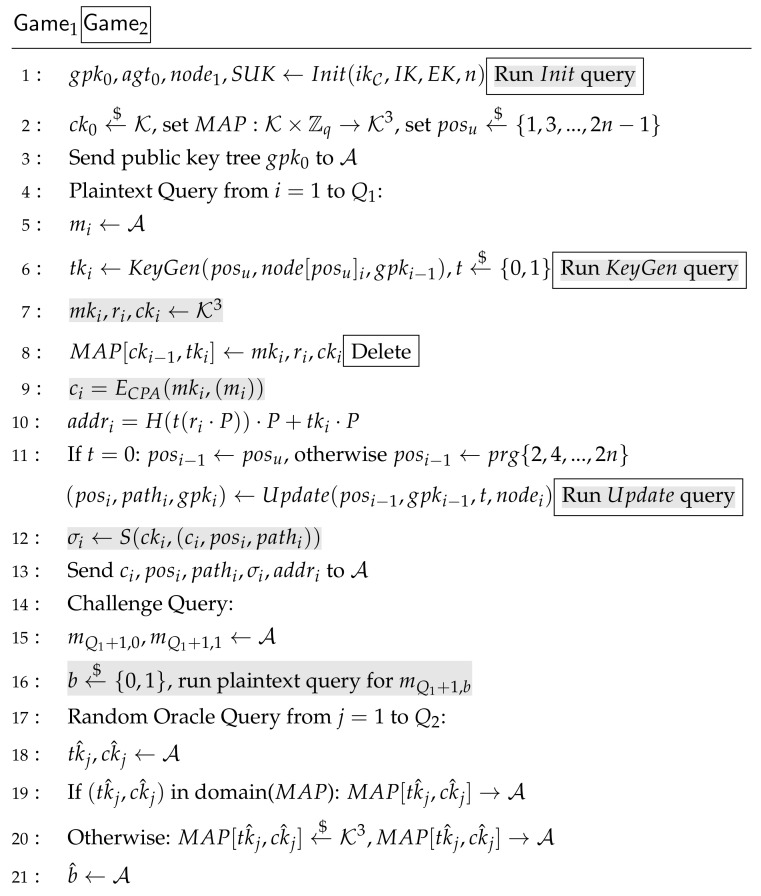
Game${}_{1}$ Challenger and Game${}_{2}$ Challenger for IND-CPA.

**Figure 8 sensors-21-01058-f008:**
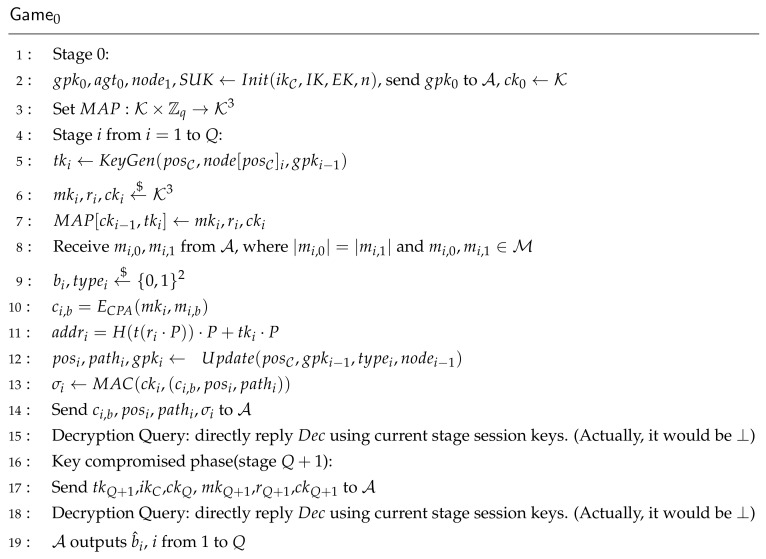
Game${}_{0}$ Challenger for FS.

**Figure 9 sensors-21-01058-f009:**
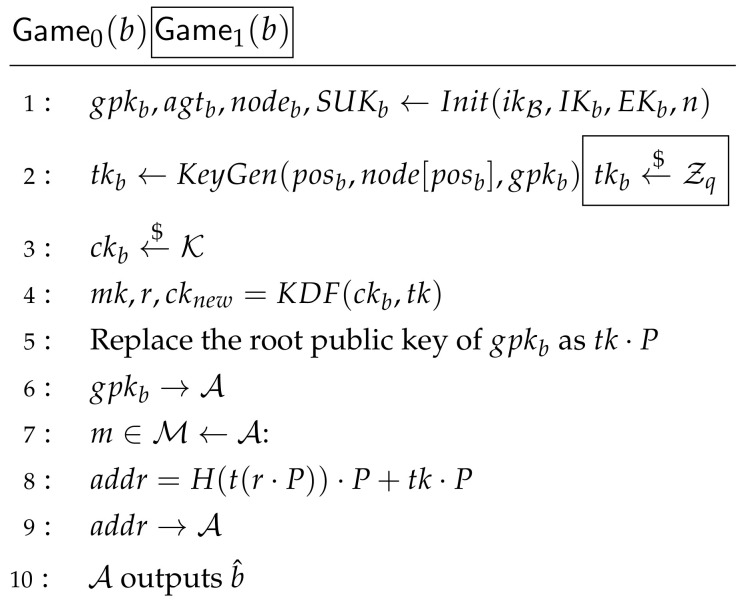
Game${}_{0}$ and Game${}_{1}$ Challengers for EGA.

**Table 1 sensors-21-01058-t001:** Performance comparison. *n* denotes the group size, each key exchange operation will access the exponentiation one time. Each key exchange, exponentiation, and encryption will cost one storage.

		#Exponentiation Times		#Encryption Times		#Communication Storage		#Computation Storage	
		Sender	Per Other	Sender	Per Other	Sender	Per Other	Sender	Per Other
sender keys	setup	*n*	*n*	*n*	*n*	*n*	*n*	2n	2n
	ongoing	0	0	1	1	1	1	1	1
pair-wise Signal	setup	*n*	*n*	0	0	*n*	*n*	*n*	*n*
	ongoing	*n*	1	n−1	1	*n*	1	2n	2
ART	setup	2n	log(n)	0	0	3n−1	3n−1	2n	log(n)
	ongoing	log(n)	log(n)	1	1	log(n)+1	log(n)+1	log(n)+1	log(n)+1
Ours	setup	4n	log(2n)	0	0	5n−1	4n+1	4n	log(2n)
	ongoing	log(2n)	log(2n)	1	1	log(2n)+1	log(2n)+1	log(2n)+1	log(2n)+1

**Table 2 sensors-21-01058-t002:** Security comparison.

Apps or Protocols	E2EE	FS	PCS	EGA	IGA
iMessage	Yes	Yes	No	No	No
LINE	Yes	No	No	No	No
Signal	Yes	Yes	No	No	No
ART	Yes	Yes	Yes	No	No
Tor and ToK	No	No	No	Yes	Yes
KEM/DEM	Yes	No	No	Yes	No
QQ and WeChat	Unknown	Yes	No	Unknown	Unknown
Ours AART	Yes	Yes	Yes	Yes	Yes

## Data Availability

Not applicable.

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
