# Peer review of "Anonymous Asynchronous Ratchet Tree Protocol for Group Messagingâ€"

_sensors, 2021, doi:10.3390/s21041058_

Round 1
Reviewer 1 Report
The paper extends the Asynchronous Ratchet Tree (ART) Protocol with anonymity notions. The goal is to achieve internal and external group anonymity. The former tries to achieve anonymity even if the adversary is part of the group. The latter provides anonymity with respect to an adversary that is not part of the group. The main idea is to double the size of the ART and extend it with Monero-style one-time addresses that provide anonymity. The submitted extends the version published at ICISC 2020 with additional proofs.
The topic is highly interesting, but the paper lacks in presentation and some technical contributions are not clear. In particular, the syntax and security model of an anonymous ART needs to be improved:
In Section 3 it is unclear what the expected use of the algorithms is. In particular, some algorithms such as the Update and UpdateGpk algorithms are introduced without discussing their purpose. If one is familiar with forward secure cryptosystems, one might guess what Update is supposed to do. At the same time, calling of the Update algorithm is missing from the FS and PCS security experiments. UpdateGpk is not used at all in the security experiments. I would recommend to restructure Section 3 as starting with the syntax of the anonymous ART protocol, a correctness definition and then the introduction of the security notions.
As the internal and external group anonymity notions are introduced in this paper (and the conference version of the paper, respectively), I would expect discussion on the intuition of these two notion, i.e., the definitions should include some discussion how those two notions achieve the desired properties. I am also a bit surprised that only IND-CPA-style security notions are considered. In particular, as algorithms for MACs are given as part of the system. What would happen if Decryption oracles would be added to the security experiments? In particular, an IND-CCA style notion for internal group anonymity, forward security and post-compromise security would be important. An adversary may have access to old/new ciphertexts in this cases which it might be able to decrypt.
The construction presented by the authors requires a trusted third party to set up the tree structure. The paper however lacks a discussion on this trusted third party (TTP). For a group messaging protocol and given what Signal does, a trusted third party seems undesirable. What are the consequences of having a TTP in this system?
In the discussion on WeChat and QQ it is not clear between which endpoints the TLS connections is established.
The paper definitely needs a pass with a spelling and grammar checker. These issues include:
* Ixternal -> Internal
* Diffi -> Diffie
* Hallman -> Hellman
* KeyEachange -> KeyExchange
* acording -> according
* reducted -> reduced
* seuciry -> security
* comparation -> comparison
* it can be proved -> it can be proven
* by one-to-one secure protocol -> with a one-to-one secure protocol
* this strategy is called as "sender keys" -> this strategy is called "sender keys"
* lead to amount of user information leaked - what amount?
* thus, user identity may be -> thus, a user's identity may be
* , We -> , we
* missing space before (DH) in 2.1.3
* to message server -> to the message server
* pair-wised?
* to fresh -> to refresh
* are most popular M applications -> are the most popular IM applications
* sent to proxy, and proxy delays -> sent to a/the proxy, and the proxy delays
* the identity of sender can be -> the identity of senders can be
I stopped checking after Section 2.
Author Response
We have revised our manuscript according to your review report, please see the attachment. Thank you for your review and advice.

Reviewer 2 Report
The paper presents a multi-stage anonymous group messaging protocol based on the asynchronous ratchet tree. The paper s generally well written and presents an interesting idea related to protocols which offer anonymity and other security features (e.g. PFS). However, it needs to be improved in the following aspects:
- In the discussion, the security features and a security comparison to related schemes need to be added.
- The performance analysis and comparison could be supplemented by storage considerations as currently computation and communication costs are included.
- A reference implementation would be beneficial or some simulation results in order to demonstrate the applicability of the presented scheme.
Author Response

(The authors gave the same response as above.)

Round 2
Reviewer 2 Report
The revised version of the manuscript all the issues raised by the reviewer.